# AUDITING PRIVACY PROTECTION OF MACHINE UN-LEARNING

## ABSTRACT

Machine unlearning aims to remove the effect of specific data from trained models to ensure individuals' privacy. However, it's arguable how to evaluate whether the privacy protection goal is achieved by machine unlearning. Furthermore, recent studies show unlearning may also increase the retained samples' privacy risks. This paper takes a holistic approach to auditing both unlearned and retained samples' privacy risks before and after unlearning. We derive the privacy criteria for unlearned and retained samples, respectively, based on the perspectives of differential privacy and membership inference attacks. To make the auditing practical, we also develop an efficient membership inference attack, A-LiRA, utilizing data augmentation to reduce the cost of shadow model training. Our experimental findings indicate that existing machine unlearning algorithms do not consistently protect the privacy of unlearned samples and may inadvertently compromise the privacy of retained samples. For reproducibility, we have pubished our code.[1]

## 1 INTRODUCTION

The *"right to be forgotten"* allows data contributors to request the deletion of their data from an organization's records. Recent regulations, including the General Data Protection Regulation (GDPR) in the European Union, the California Consumer Privacy Act (CCPA) in the U.S., and Canada's Personal Information Protection and Electronic Documents Act (PIPEDA), have solidified this right. For example, Google received over 3.2 million requests to remove specific URLs from search results over five years, demonstrating the importance and scale of this issue.

In the context of *machine learning*, exercising the right to be forgotten entails not only removing the data itself but also ensuring models are retrained with the updated dataset. However, with the rise of large models, frequent model retraining brings unbearable costs to model owners. Thus, the process of efficiently adjusting models to avoid high retraining costs, known as *machine unlearning*, has garnered significant attention in both academic and industrial research. Most recent machine unlearning methods (Golatkar et al. (2020); Foster et al. (2024); Fan et al. (2024)) claim that machine unlearning can be a promising way to protect privacy, yet the effectiveness of protection has not been sufficiently evaluated. A popular argument is that unlearning is effective in privacy protection as long as any impact the erased data had on the original model is removed from the unlearned model, which is known as the *completeness* measure. However, Chen et al. (Chen et al. (2021)) highlight that current machine unlearning techniques may still leave residual information about the target data. More importantly, is the completeness measure appropriate for privacy protection?

Moreover, Carlini et al. (Carlini et al. (2022b)) note that a sample's privacy is a *relative* notion, meaning that removing some samples could inadvertently increase the privacy risks of other samples in the retained data. It's consistent with the understanding of anonymization by *blending in the crowd* — a thinning crowd leaves less protection to the crowd members. This issue is particularly concerning when the model builder has promised to preserve the entire dataset's privacy with a certain guarantee level, i.e., the $\epsilon$ setting, in differentially private machine learning (Dwork (2006); Abadi et al. (2016)).

---

[1] https://anonymous.4open.science/r/Auditing-machine-unlearning-CB10/README.md

**Scope of Research**. We take a holistic approach to measure the impact of a machine unlearning algorithm on both the unlearned and retained samples. Specifically, we consider successful machine unlearning in privacy protection should meet the following goals: (1) the privacy risk of unlearned samples should decrease as much as possible after unlearning, and (2) the privacy risk of retained samples should remain below the promised bound if a differentially private algorithm was applied in modeling.

Recent studies on auditing differentially private machine learning algorithms give some ideas for us to analyze and define the above two goals formally. The auditing mechanism uses the theorem (Dwork et al. (2014); Tramer et al. (2022); Jagielski et al. (2020)) that if a $\epsilon$-differentially-private machine learning algorithm is correctly implemented, the worst-case membership inference (MIA) test result on *every sample* must satisfy $\ln(\text{TPR}/\text{FPR}) < \epsilon$, where TPR and FPR are the true positive and false positive rates of the membership inference test, respectively. To make auditing effective, the MIA test must be effective, e.g., $\ln(\text{TPR}/\text{FPR})$ is as close to its theoretical upper bound as possible. So far, the LiRA method (Carlini et al. (2022a)) is considered the best MIA method that is able to measure individual sample's $\ln(\text{TPR}/\text{FPR})$ more accurately than other methods (Carlini et al. (2022b)). With a sample-level privacy risk measure, we can formally define the two privacy protection goals for machine unlearning methods.

A major challenge in applying the LiRA method is its high computational cost. Online-LiRA requires 4.8 GPU hours per sample to generate the TPR/FPR measure. While offline-LiRA is more efficient due to its one-sided hypothesis testing, its performance is inferior to that of online-LiRA. In this paper, we introduce a more efficient augmentation-based likelihood ratio attack (A-LiRA), which achieves comparable performance to online-LiRA with a 88.3% reduction in time cost. A-LiRA is inspired by Mattern et al. (2023) and leverages the central limit theorem. Instead of training multiple shadow models as in online-LiRA, A-LiRA approximates the two probability distributions – when the model is trained with or without the target sample – using augmented data and with the assumption of normal distributions for in- and out-training measures, separately. This significantly reduces the time cost while providing a good approximation of the probability distributions, making A-LiRA both effective and efficient for estimating the privacy risk of a sample.

We have conducted extensive evaluations to demonstrate that A-LiRA provides comparable effectiveness while significantly improved efficiency compared to the online-LiRA approach introduced by Carlini et al. (2022b). We also show that most recent machine unlearning methods do not satisfactorily protect the privacy of unlearned samples and, in some cases, increase the privacy risk of retained samples. Thus, new unlearning methods are desired to meet the proposed privacy measures. In summary, our contributions are as follows:

- We re-formulate the criteria for privacy protection of machine unlearning methods based on the theories of differential privacy and membership inference attacks.

- We introduce a novel augmentation-based likelihood-ratio attack to estimate the privacy risk of samples, which allows us to calculate the privacy protection criteria efficiently.

- We show that most existing machine unlearning methods do not protect privacy well for the unlearned samples, and in some cases, they may also increase privacy risks for the retained samples.

The remaining sections are organized as follows: Related Works (Section 2), Preliminaries(Section 3), Privacy Risk Measures for Machine Unlearning (Section 4), Augmentation-Based Likelihood Ratio Attack (Section 5), and Experiments (Section

## 2 RELATED WORKS

Machine unlearning enforces the "right to be forgotten" and is categorized into exact and approximate methods. Exact unlearning retrains models without the forgotten data. A notable example is SISA by Bourtoule et al. (2021), which partitions training data into micro-shards and trains small models on each. To unlearn data, only the relevant micro-models are retrained, improving efficiency over full retraining. Approximate unlearning, on the other hand, employs techniques such as weight manipulation and fine-tuning to make the model "forget" the data without retraining it entirely. For instance, Golatkar et al. (2020) modifies the weights to make probing functions of the

weights indistinguishable from those applied to a network trained without the data being forgotten. Fan et al. (2024) introduces the concept of 'weight saliency' for machine unlearning, drawing parallels with input saliency used in model explanation. This approach focuses unlearning efforts on specific model weights, improving both efficiency and effectiveness. While machine unlearning is touted as a means to protect privacy (Cao & Yang (2015); Nguyen et al. (2022); Xu et al. (2023)), studies such as those by Chen et al. (2021) and Carlini et al. (2022b) indicate that unlearning may make retained samples more susceptible to membership inference attacks. These findings highlight the necessity for robust auditing methods and standards to ensure the privacy protection efficacy of machine unlearning.

## 3 PRELIMINERIES

In this section, we introduce some preliminaries to provide the audience with a better understanding of this paper.

### 3.1 NOTATIONS

We denote the training dataset as $D$ and the subset to be unlearned as $X$. The retained samples are represented by $D \setminus X$. The model trained on $D$ using algorithm $\mathcal{M}$ is denoted as $M = \mathcal{M}(D)$, while $U = \mathcal{U}(\mathcal{M}(D), X)$ represents the unlearned model after applying the unlearning mechanism $\mathcal{U}$ on $X$. The model retrained on the retained dataset $D \setminus X$ is expressed as $M^R = \mathcal{M}(D \setminus X)$. For membership inference attacks (MIA), $MIA(U, D, x)$ indicates an MIA applied to the unlearned model $U$ for a target sample $x$. Lastly, $E(M, x)$ refers to the privacy risk estimator of sample $x$ in model $M$.

### 3.2 DIFFERENTIAL PRIVACY

Differential Privacy (DP) ensures that the inclusion or exclusion of a single data point minimally affects the output of an algorithm, protecting individual privacy. A machine learning algorithm $\mathcal{M}$ satisfies $(\epsilon, \delta)$-DP if, for any two neighboring datasets $D$ and $D'$ that differ in at most one sample, and any output set $S$:

$$\Pr[\mathcal{M}(D) \in S] \le e^\epsilon \cdot \Pr[\mathcal{M}(D') \in S] + \delta$$

Here, $\epsilon$ (privacy loss) controls the strength of the guarantee, with smaller $\epsilon$ offering stronger privacy. $\delta$ allows for a small probability of privacy violation, especially for extreme outputs. Lower values of $\epsilon$ and $\delta$ indicate higher privacy protection.

### 3.3 LIKELIHOOD RATIO ATTACK

Likelihood ratio attacks (LiRA) were first introduced by Carlini et al. (2022a) for machine learning models. Given a target sample $x$, the online-LiRA sets up two hypotheses:

$$H_0 : \text{Target model is trained on } D,$$
$$H_1 : \text{Target model is trained on } D \setminus x$$

It trains multiple shadow models (typically 512 in total) on $D$ or $D \setminus x$ fits the logit outputs to Gaussian distributions. The likelihood ratio between the two distributions is used to reject one of the hypotheses. While effective, this method is computationally expensive, taking an average of 4.8 hours per sample in CIFAR-10 on a Titan V100 GPU, making it impractical. A more efficient variant, offline-LiRA, only estimates the distribution $\mathcal{D}(x|\mathcal{M}(D \setminus x))$ of the target model is not trained on the target model and uses the likelihood of $x$ follows the estimated distribution to reject or accept $H_1$. This one-sided approach can be batched by training shadow models on random subsets of $D$, but it performs worse than the online-LiRA in efficacy.

## 4 A HOLISTIC MEASURE FOR AUDITING MACHINE UNLEARNING

### 4.1 THREAT MODEL

Before introducing the measure for auditing machine unlearning, we outline the threat model for our proposed attack.

**Target Model and Sample.** The target model consists of an original model, denoted as $\mathcal{M}(D)$, trained on dataset $D$, and an unlearned model, $U = \mathcal{U}(\mathcal{M}(D), X|X \subset D)$, where $\mathcal{U}$ represents the unlearning mechanism applied to a sample set $X$. According to the ideal definition of machine unlearning (Xu et al. (2023); Nguyen et al. (2022)), the unlearning method should be equivalent to the model retrained on the retained set $D \setminus X$, such that $U = M^R$, where $M^R = \mathcal{M}(D \setminus X)$. The model builder may have an agreement with data contributors that regulates the privacy risk of the data, which data owners won't express at a certain level.

**Attacker's Capabilities.** To assess the privacy risks associated with individual samples, we consider a worst-case adversary. The attacker has white-box access to both the original model $\mathcal{M}(D)$ and the unlearned model $\mathcal{U}(\mathcal{M}(D), X|X \subset D)$, as well as full knowledge of the training dataset $D$ and the unlearned samples $X$. Additionally, the attacker knows the details of the unlearning mechanism $\mathcal{U}$.

**Attacker's Objective.** The attacker's goal is to determine whether a given sample from $D$ was used to train the target models. Ideally, a successful attack would classify any sample in $D$ as a "member" in the original model $\mathcal{M}(D)$ and any retained sample (i.e., $x \in D \setminus X$) as a member in the unlearned model $\mathcal{U}(\mathcal{M}(D), X|X \subset D)$. Furthermore, for each unlearned sample $x \in X$, the attacker should identify it as a "non-member" in $\mathcal{U}(\mathcal{M}(D), X|X \subset D)$. In practice, such ideal membership inference does not exist.

**Success of Defense.** Thus, we define the success of the defense as the attacker's MIA ability on the unlearned model for any sample $x \in D$, denoted as $MIA(U, D, x)$, should be equivalent or worse than that on the retrained model, $MIA(M^R, D, x)$. This criterion will be carefully studied, and an approximate version will be formally defined in later sections.

### 4.2 Auditing Machine Unlearning

Interestingly, so far, all existing machine unlearning algorithms assume we only need to ensure privacy protection for the users who execute "the right to be forgotten", and the remaining ones' privacy protection is fine to be omitted in unlearning. This assumption is incorrect if the original model is differentially private, where the model builder has reached an agreement of privacy guarantee, i.e., the $\epsilon$ setting, with data contributors. Carlini et al. (2022b) have shown the "privacy onion effect", i.e., the removal of some samples may affect other samples' privacy risks. Thus, we take a holistic approach to measure *all samples' privacy guarantees* before and after unlearning by defining following criteria:

**Criterion 1: For the unlearned samples** $X$, will unlearned $X$ adequately protect its privacy? Formally, if we have an ideal estimator $E$ to estimate privacy risk and the unlearning algorithm works ideally, for every sample $x \in X$, we have

$$E(\mathcal{U}(\mathcal{M}(D), X), x|x \in X) \approx E(\mathcal{M}(D/X), x|x \in X)$$

This equation indicates that the privacy risk of unlearned samples on the unlearned model should approximate the privacy risk of a retrained model. In practice, without knowing the retrained model $\mathcal{M}(D/X)$, the following inequality *must* hold for an effective unlearning algorithm:

$$E(\mathcal{U}(\mathcal{M}(D), X), x|x \in X) < E(\mathcal{M}(D), x|x \in X) - t_1, \quad t_1 \geq 0, \tag{1}$$

where $t_1$ is a pre-defined threshold that model builders use to achieve stricter or relaxed privacy guarantees. When $t_1 = 0$, the equation reflects the minimum requirement on the unlearning algorithm. That is, the privacy risk of unlearned samples on unlearned models should be less than that on the original model. Violating Equation 1 at $t_1 = 0$ implies the unlearning algorithm does not achieve the goal of privacy protection for certain samples.

**Criterion 2: For the retained samples** $D \setminus X$, will unlearned $X$ increase the privacy risk of the retained data and possibly break a preset $\epsilon$ value in differentially private modeling? Let $F(\epsilon)$ be the upper bound of the estimator determined by $\epsilon$. Since the original algorithm is differentially private, we have

$$E(\mathcal{M}(D), x|x \in D) \leq F(\epsilon)$$

Specifically, when the risk estimator uses $\ln(TPR/FPR)$, $F(\epsilon) = \epsilon$ (Tramer et al. (2022); Jagielski et al. (2020)).

With machine unlearning, ideally, we also have to meet the privacy guarantee

$$E(\mathcal{U}(\mathcal{M}(D), X), x | x \in D \setminus X) \leq F(\epsilon).$$

In practice, to strictly ensure the privacy guarantee is still met and avoid deriving the form of $F(\epsilon)$, we consider a reasonable machine unlearning should meet the following conditions:

$$E(\mathcal{U}(\mathcal{M}(D), X), x | x \in D \setminus X) \leq \max(E(\mathcal{M}(D), x | x \in D)) + t_2, \quad (2)$$

where $t_2$ defines the relaxation level. At $t_2 = 0$, if Equation 2 does not hold at certain samples, the privacy guarantee *may* still be satisfied unless $E(\mathcal{U}(\mathcal{M}(D), X), x \in D \setminus X) > F(\epsilon)$ is also true. This indicates that the privacy risk of retained samples on unlearned models should not be greater than the greatest privacy risk among every sample on the original model. Based on these two criteria, we introduce an *acceptance matrix* (Figure 1) that determines whether a machine unlearning method is applicable to protect privacy: for a differentially private model, the method is applicable only when it satisfies both Equation 1 and 3. For a plain model, the method must satisfy Equation 1. The need for satisfaction of Equation 2 is decided by the data contributors.

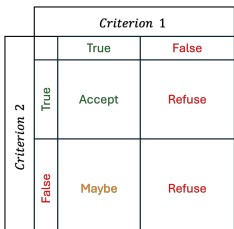

### 4.3 PRIVACY RISK ESTIMATOR $E$

Figure 1: Acceptance matrix

To implement the above auditing method, we need an accurate and efficient estimator $E$ to estimate the privacy risk at the sample level. This estimator must be theoretically sound and computationally efficient. Dwork et al. (2014) interpreted differential privacy from the perspective of hypothesis testing, where they proved that if an algorithm is $(\epsilon, \delta)$-differentially private, then for any distinguishing attack (e.g., membership inference attacks), the ratio of the true positive rate (TPR) to the false positive rate (FPR) satisfies $\text{TPR}/\text{FPR} < e^\epsilon$. Thus, we can take the following estimator to define privacy risk:

$$E(M, x) = \ln \left( \frac{TPR(\mathcal{A}, M, x)}{FPR(\mathcal{A}, M, x)} \right)$$

where TPR and FPR are the corresponding MIA attack $\mathcal{A}$ applied to the model $M$ targeting the sample $x$.

When $M = \mathcal{M}(D)$ is $(\epsilon, \delta)$-differentially private and an unlearned model is $U = \mathcal{U}(\mathcal{M}(D), X)$, it's straightforward to have $\ln(\text{TPR}/\text{FPR}) < \epsilon$. Equation 2 is simplified to

$$\ln \left( \frac{\text{TPR}(U, x)}{\text{FPR}(U, x)} \right) \leq \max \left( \ln \left( \frac{\text{TPR}(M, x)}{\text{FPR}(M, x)} \right) \right) + t_2 < \epsilon, \text{ where } x \in D \setminus X. \quad (3)$$

Model builders can adjust $t_2$ within the range $[0, \epsilon - \max(\text{TPR}(M, x)/\text{FPR}(M, x)))$ to determine the relaxation level without violating differential privacy. When $\ln(TPR(U, x)/FPR(U, x)) \geq \epsilon$ is true for some $x$, we can say that machine unlearning breaks differential privacy.

When the $\mathcal{M}(D)$ is not trained with the differential privacy, we can still detect the samples with $\ln(TPR(U, x)/FPR(U, x)) > \max(\ln(TPR(M, x)/FPR(M, x)))$ for the increased privacy risks.

## 5 AUGMENTATION-BASED LIKELIHOOD RATIO ATTACK (A-LIRA)

Given the estimator $E(\cdot)$, we need a membership inference attack $\mathcal{A}$ to classify whether a target sample $x$ was used to train a model. For an unlearned model $U = \mathcal{U}(\mathcal{M}(D), X | X \subset D)$, $\mathcal{A}$ should classify $x \in X$ as a "non-member" and $x \in D \setminus X$ as a "member."

The attack $\mathcal{A}$ must be powerful to maximize the TPR/FPR ratio and maintain high TPR at low FPR, which is critical for security analysis (Kolter & Maloof (2006); Kantchelian et al. (2015); Carlini et al. (2022a)). LiRA (Carlini et al. (2022a)) is the most accurate MIA algorithm in the low-FPR region, but online-LiRA is costly, while offline-LiRA sacrifices accuracy. The performance bottleneck

of online-LiRA stems from training many shadow models. Online-LiRA trains $2n$ shadow models to fit two distributions ($n = 256$ typically), costing 4.8 GPU hours for CIFAR datasets. Inspired by Mattern et al. (2023), we propose A-LiRA, an augmentation-based LiRA algorithm. Instead of training $2n$ shadow models, A-LiRA only trains one in-training shadow model $\mathcal{M}(D|x \in D)$ and one out-training shadow model $\mathcal{M}(D \setminus x)$, applying $n$ augmented samples to observe the output distributions. A-LiRA is detailed in Algorithm 1 and operates in the following three phases.

---

**Algorithm 1** Augmentation-based LiRA

---

**Require:** target model $f_T$, example $(x, y)$, dataset $D$, number of augmentations $n$
  1: $\text{obs}_{\text{in}} \leftarrow \{\}$
  2: $\text{obs}_{\text{out}} \leftarrow \{\}$
  3: $\text{obs}_{\text{target}} \leftarrow \{\}$
  4: $f_{\text{in}} \leftarrow \mathcal{M}(D)$                                      ▷ Train shadow IN model
  5: $f_{\text{out}} \leftarrow \mathcal{M}(D \setminus \{(x, y)\})$                 ▷ Train shadow OUT model
  6: $X_{\text{aug}} \leftarrow \text{Augmentation}(x, n)$                      ▷ Augment the data
  7: **for** $x_i \in X_{\text{aug}}$ **do**
  8:     $\text{obs}_{\text{in}} \leftarrow \text{obs}_{\text{in}} \cup \{\phi(f_{\text{in}}(x_i)_y)\}$
  9:     $\text{obs}_{\text{out}} \leftarrow \text{obs}_{\text{out}} \cup \{\phi(f_{\text{out}}(x_i)_y)\}$
10:     $\text{obs}_{\text{target}} \leftarrow \text{obs}_{\text{target}} \cup \{\phi(f_T(x_i)_y)\}$
11: **end for**
12: $\mu_{\text{in}} \leftarrow \text{mean}(\text{obs}_{\text{in}})$
13: $\mu_{\text{out}} \leftarrow \text{mean}(\text{obs}_{\text{out}})$
14: $\sigma_{\text{in}}^2 \leftarrow \text{var}(\text{obs}_{\text{in}})$
15: $\sigma_{\text{out}}^2 \leftarrow \text{var}(\text{obs}_{\text{out}})$
16: Determine $\tau$ through thresholding.
17: $\Lambda \leftarrow \dfrac{PDF(\max(\text{obs}_{\text{target}})|\mathcal{N}(\mu_{\text{in}}, \sigma_{\text{in}}^2))}{PDF(\max(\text{obs}_{\text{target}})|\mathcal{N}(\mu_{\text{out}}, \sigma_{\text{out}}^2))}$
18: **return** "member" if $\Lambda > \tau$ else "non-member"

---

**Phase 1: Distribution estimation.** For a target sample $(x, y)$, we generate $n$ augmentations $X_{\text{aug}}$ by randomly flipping, rotating, and shifting $x$. We then train two shadow models: an *in*-model $f_{\text{in}} = \mathcal{M}(D)$ (trained with $x$ in the dataset) and an *out*-model $f_{\text{out}} = \mathcal{M}(D \setminus x)$.

For each augmented sample $x_i \in X_{\text{aug}}$, we input it into each of the two models $f_{\text{in}}$ and $f_{\text{out}}$ and obtain the confidence vector $l_i$, which contains $c$ elements for a $c$-class prediction, with $\sum_c l_{i,c} = 1$. We focus on the confidence of the true label $y$, denoted as $l_{i,y}$ for distribution estimation. Following Carlini et al. (2022a), we apply a logit transformation to $l_{i,y}$:

$$\phi(p) = \log\left(\frac{p}{1-p}\right), \quad \text{where } p = l_{i,y}.$$

The transformed confidence values are assumed to follow two normal distributions $\mathcal{N}(\mu_{\text{in or out}}, \sigma_{\text{in or out}}^2)$, with parameters estimated from the collection of $\{\phi(l_{i,y})\}$. Typically, $f_{\text{in}}$ will have a higher mean and smaller variance. The output of this phase is the two normal distributions: $\mathcal{N}(\mu_{\text{in}}, \sigma_{\text{in}}^2)$ and $N(\mu_{\text{out}}, \sigma_{\text{out}}^2)$.

**Phase 2: Making decision.** To determine whether $x$ was used to train the target model $f_T = \mathcal{M}(D)$, the augmented set $X_{\text{aug}}$ is passed through the model, and the same logit transformation is applied to the output confidence values. Intuitively, if $f_T$ was trained on $x$, the transformed confidence values $\phi(l_{i,y})$ should vary less compared to when $f_T$ was not trained on $x$. The maximum value of $\phi(l_{i,y})$ highlights the largest difference between these two cases. We then compute the likelihood of $\max\{\phi(l_{i,y})\}$ follows either of the two normal distributions $\mathcal{N}(\mu_{\text{in}}, \sigma_{\text{in}}^2)$ or $\mathcal{N}(\mu_{\text{out}}, \sigma_{\text{out}}^2)$. As shown by Carlini et al. (2022a), the best true positive rate (TPR) at a given false positive rate (FPR) can be found by thresholding the likelihood ratio $\Lambda$:

$$\Lambda = \frac{PDF(\max\{\phi(l_{i,y})\}|\mathcal{N}(\mu_{\text{in}}, \sigma_{\text{in}}^2))}{PDF(\max\{\phi(l_{i,y})\}|\mathcal{N}(\mu_{\text{out}}, \sigma_{\text{out}}^2))}.$$

where PDF is the probability density function. This phase outputs the computed $\Lambda$ for the target model $f_T$ and sample $x$. Assuming we have an ideal threshold $\tau$, if $\Lambda > \tau$, we conclude that $f_T$ was trained on $x$; otherwise, it was not.

**Phase 3: Generating threshold $\tau$ at a specific FPR.** Due to the page limit, we introduce Line 16 in Algorithm 1 here. To determine the optimal threshold $\tau$ for the best TPR at a given FPR, we follow the method from Carlini et al. (2022a). We train $m$ *in*-models $f_{\text{in}}^j = \mathcal{M}(D)$ and $m$ *out*-models $f_{\text{out}}^j = \mathcal{M}(D \setminus x)$, where $j = \{1, \ldots, m\}$. Each model is treated as a target model, with *in*-models labeled as "member" and *out*-models as "non-member."

For each model, we calculate the likelihood ratio $\Lambda_{\text{in or out}}^j$ using the normal distributions from the distribution estimation phase. This results in $2m$ likelihood ratios $\{\Lambda_{\text{in or out}}^j\}$. A model with $\Lambda > \tau$ is classified as a "member," otherwise as a "non-member." By testing $\tau$ values from $\{\Lambda_{\text{in or out}}^j\}$, we select the threshold that achieves the best TPR at the given FPR.

**Time cost.** In our work, we set $m = 30$ for both A-LiRA and online-LiRA. Thus our A-LiRA needs to train $2 + 2m$ models while online-LiRA needs to train $2n + 2m$ models, which shows the improved efficiency given by augmentation. When $n = 256$ and $m = 30$, A-LiRA will reduce the time cost by 89%, which aligns with our experimental results.

**Estimation of privacy risk.** We generate $k$ unlearned models $U_t = \mathcal{U}(\mathcal{M}(D), X | X \subset D)$, where $t = 1 \cdots k$ and $X$ represents the forgotten samples.

For Criterion 1, we apply A-LiRA with the corresponding distributions and threshold $\tau$ for each target sample $x \in X$. Using each $U_t$ as the target model, we decide if $x$ is a "member" or "non-member." Comparing these results with the true status (all "non-member"), we compute the privacy risk as $\ln(\text{TPR}/\text{FPR})$. A higher value indicates a greater privacy risk for $X$.

For Criterion 2, we apply the same process using retained samples $x \in D \setminus X$, where the true status is "member." In our experiments, we set $k = 30$ for efficiency.

# 6 EXPERIMENTS

This section presents extensive experiments aimed at addressing the following key questions: 1. How does A-LiRA perform in terms of both efficacy and efficiency compared to the LiRA methods? 2. How well do the published unlearning methods meet the proposed auditing measures?

## 6.1 SETUP

Before exploring these questions, the experimental setup is outlined.

**Datasets and models:** We use CIFAR-10 and CIFAR-100 to train ResNet-18 models with FFCV (Leclerc et al. (2023)), using a learning rate of 0.5, weight decay of 5e-4, and training for 50 epochs with early stopping on NVIDIA Titan V-100 and Titan Xp GPUs. Each sample has 10 random augmentations to enhance generalization. Due to computational constraints, we limit our experiments to these datasets, as large-scale datasets would require training multiple models for privacy risk estimation, which is impractical with current resources.

**Membership inference attacks:** We use A-LiRA as the attack method, generating 1000 augmentations per sample via random flipping, rotation, shifting, etc. One shadow *in*-model and one shadow *out*-model are trained to estimate the distributions. We then train 30 *in*- and *out*-models to determine the threshold. All models use the same hyperparameters and data distribution as the target model. Results of online-LiRA and offline-LiRA (Carlini et al. (2022a)) are provided in Appendix A.

**Metric.** We use the Area Under the ROC Curve (AUC) score, TPR when FPR $= 0.01$, and GPU time to evaluate membership inference attacks. We use the ratio of samples that failed to meet Criterion 1&2 to evaluate the privacy protection of machine unlearning methods. Specifically, we check the ratio of unlearned samples that do not meet Criterion 1 and the ratio of retained samples that do not meet Criterion 2, denoted as the "Failure Rate" in the figures.

## 6.2 ATTACKING PERFORMANCE

We present the attack performance in Table 1. All models are trained using ResNet-18. For both online and offline-LiRA, we totally train 512 shadow models (in and out), which is suggested by Carlini et al. (2022a). In the offline-LiRA setting, we use a batch size of 50, meaning that 50 samples

| Method | CIFAR-10 | | | CIFAR-100 | | |
|---|---|---|---|---|---|---|
| | AUC | TPR@1%FPR | Time(hours) | AUC | TPR@1%FPR | Time(hours) |
| Online-LiRA | **0.706** | 9.7% | 4.76 | 0.913 | 25.4% | 4.84 |
| Offline-LiRA | 0.663 | 8.6% | **0.17** | 0.833 | 19.3% | **0.13** |
| A-LiRA(Ours) | 0.703 | **10.3%** | 0.53 | **0.917** | **26.8%** | 0.57 |

Table 1: Attacking models performance on CIFAR datasets. TPR@1%FPR indicates TPR when FPR is 0.01. Time is the GPU hours to generate classificaiton for one sample, including training shadow models and thresholding. The bolded cell is the best performance.

are explicitly used to train the out-models, and the attack is performed on those same samples. While Carlini et al. (Carlini et al. (2022b)) demonstrated that batch strategies can be applied to both online and offline-LiRA, and that setting the batch size to half the dataset size (i.e., half the data used to train the shadow *in*-model) can be effective, we found this approach reduced attack effectiveness.

Online-LiRA performed the best in global AUC, while our A-LiRA achieved comparable AUC and better TPR at low FPR with significantly lower time costs. On CIFAR-100, A-LiRA outperformed in both AUC and TPR at low FPR. Although offline-LiRA yielded the lowest performance across all experiments, it substantially reduced time costs, making it a viable alternative when time efficiency is a priority.

### 6.3 DOES UNLEARNING PROTECT PRIVACY?

In this section, we investigate whether existing machine unlearning satisfactorily protects privacy according to the proposed criteria.

#### 6.3.1 MODEL RETRAINING

We start with how directly retained models perform on these criteria. Specifically, we first estimate the privacy risk of each sample in $D$ on model $\mathcal{M}(D)$ with A-LiRA. Then, we remove samples with the top-$k\%$ privacy risks, retrain a new model, and evaluate the samples' privacy risk under the new model.

**Criterion 1.** Figure 2 shows the result for retraining with top-k% samples removed. On both datasets, retraining will protect the removed samples well when the number of removed samples is small. However, by removing too many high-risk samples, i.e., top 20%, the risk of some samples ever presented in the dataset can rise. This phenomenon indicates that the removed samples may mutually affect each other's privacy risk, and the privacy onion effect (Carlini et al. (2022b)) may also exist within the removed samples.

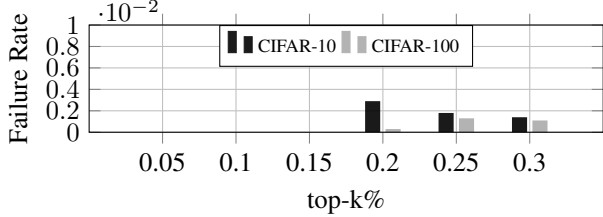

Figure 2: Criterion 1: Privacy protection under retraining.

**Criterion 2.** To evaluate how the privacy risks change on the retained samples, we first record the $\max(\ln(\text{TPR/FPR}))$ over $D$ on the original model. According to Carlini et al. (2022b), removing high-risk samples may raise other samples' privacy risk. We observe how retraining affects the retained samples by removing the top-k% and the bottom-k%-risk samples, respectively.

Figure 3 shows both unlearning top-k% and bottom-k%-risk samples will raise the privacy risk of some retained samples. However, removing low-risk samples will affect other samples less as Figure 3b shows. This observation is consistent with the previously observed "privacy onion effect"

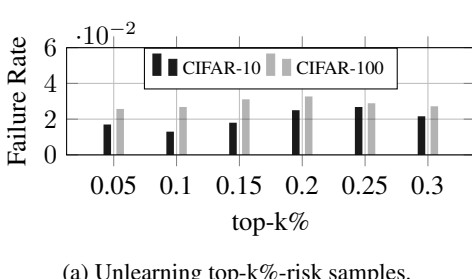

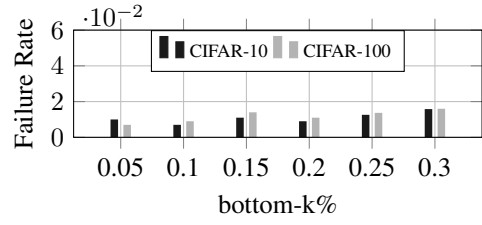

(a) Unlearning top-k%-risk samples.

(b) Unlearning bottom-k%-risk samples.

Figure 3: Criterion 2: Retraining may increase the privacy risk of some retained samples.

With these baselines for Criterion 1&2, we will study how well the three recent approximate unlearning methods protect samples' privacy.

### 6.3.2 APPROXIMATE UNLEARNING

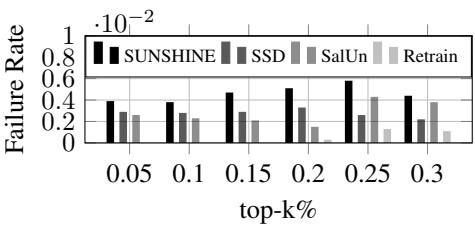

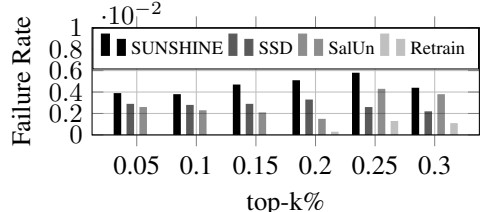

(a) Unlearning top-k%-risk samples. CIFAR-10.

(b) Unlearning top-k%-risk samples. CIFAR-100.

Figure 4: Criterion 1: Approximate unlearning delivers less privacy protection to removed samples, compared to retraining.

We choose SUNSHINE Golatkar et al. (2020), SSD Foster et al. (2024), and SalUn Fan et al. (2024) as the representative approximate unlearning methods as they are the latest algorithms with the best performance reported. Similarly, we unlearn top-k% risk and bottom-k% risk samples using each of the methods. We include the retraining result as the baseline.

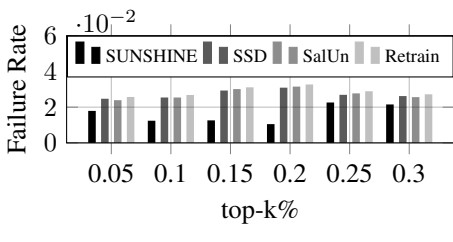

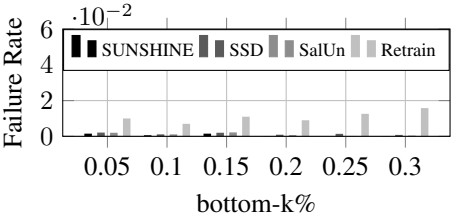

(a) Unlearning top-k%-risk samples. CIFAR-10.

(b) Unlearning bottom-k%-risk samples. CIFAR-10.

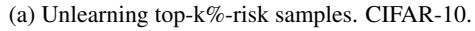

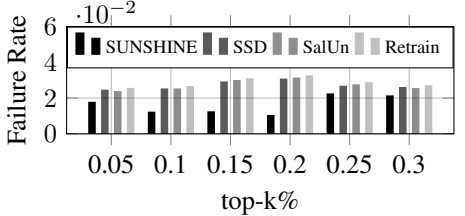

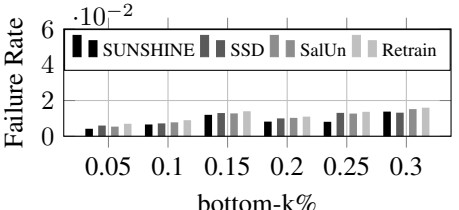

(c) Unlearning top-k%-risk samples. CIFAR-100.

(d) Unlearning bottom-k%-risk samples. CIFAR-100.

Figure 5: Criterion 2: Approximate unlearning affects less retained samples than retraining.

**Criterion 1.** The result with approximate machine unlearning is shown in Figure 4. The greater failure rates at each sample level indicate that approximate unlearning methods provide significantly less protection to unlearned samples' privacy than retraining. These results show that these approximate unlearning methods may not be satisfactory in protecting unlearned samples' privacy.

**Criterion 2.** Now we check how approximate machine unlearning affects the privacy of retained samples. Similarly, we want to observe the impact of unlearning top-k% or bottom-k%-risk samples, respectively, and check how many retained samples dissatisfy Criterion 2 (Equation 2) with $t_2 = 0$. Figure 5 shows that all three unlearning methods will increase the privacy risk of some retained samples. Interestingly, the affected samples are all less than retraining. It seems fully eliminating the unlearned samples' effect on models may invertedly increase the privacy risks of retained ones. This phenomenon is also consistent with the previously observed "privacy onion effect" (Carlini et al. (2022b)).

**Results on differentially private models.** In section 4, we've introduced that model builder and data contributor may have agreed on a privacy upper bound, e.g., the $\epsilon$ level of differential privacy if the models are differentially private. We now study the level of privacy violations approximate machine unlearning methods may bring to the retained samples.

We train the original model $\mathcal{M}(D)$ using DP-SGD introduced by Abadi et al. (2016) with various $\epsilon$ settings and keep $\delta = 0.0002$. We unlearn top-30% risk samples in CIFAR-10 using three approximate unlearning methods and check the Failure Rate of samples for Criterion 2, i.e., Equation 3 with $\epsilon$ as the bound.

Figure 6 shows that unlearning top-risk samples will cause the violation of $\epsilon$ agreement. The numbers of violated samples are significant for all $\epsilon$ levels and all the algorithms. Sunshine also affects significantly more samples at a lower $\epsilon$ level, $\epsilon = 2$. This result indicates the unlearning for differentially private algorithms should be more carefully designed to consider the protection of the retained samples.

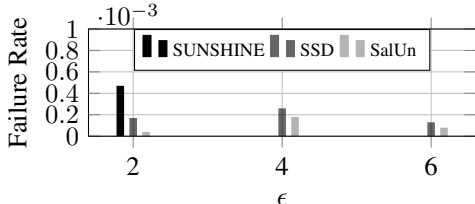

Figure 6: Approximate unlearning may break the $\epsilon$ bound for some retained samples if the models are $(\epsilon, \delta)$-differentially private.

**Discussion.** Using A-LiRA as the estimator, we have found several interesting results. Firstly, existing machine unlearning may not provide sufficient privacy protection for unlearned samples. Secondly, machine unlearning may also increase the privacy risk of retained samples, which can be critical if differentially private machine learning is applied. Thus, it's vital to reconsider how to design proper machine unlearning methods to comprehensively protect the privacy of all involved data samples. Our A-LiRA estimator and acceptance matrix can be a powerful tool to help evaluate such machine unlearning methods.

## 7 CONCLUSION

The privacy protection criteria for machine unlearning have not been sufficiently studied yet. We propose two criteria to audit the privacy risks of unlearned and retained samples to fully understand a machine unlearning algorithm's privacy protection capacity. The core of the proposed auditing mechanism is an efficient sample-level membership inference attack, A-LiRA. We show in experiments that A-LiRA performs efficiently with comparable attacking accuracy to the original online-LiRA algorithm, making it deployable to real applications. With the proposed criteria, we also show that most existing machine unlearning algorithms do not satisfactorily protect samples' privacy.

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

## A   APPENDIX A: CHOICE OF MEMBERSHIP INFERENCE ATTACK

In this section, we analyze how the choice of attack affects the estimation of privacy risks. Intuitively, in a worst-case scenario, more powerful attacks provide a better estimate of a sample's privacy risk, as stronger attacks are more likely to detect samples that fail to meet both criteria. We evaluate online-LiRA, offline-LiRA, and A-LiRA by generating $\ln(\text{TPR}/\text{FPR})$ and repeating the experiments on CIFAR-10 as described in Section 6.3.1. The results for Criterion 1 are shown in Figure 7. Both online-LiRA and A-LiRA produce comparable estimates, while offline-LiRA detects fewer samples that fail to meet Criterion 1.

Similar findings are shown in Figure 8, where we evaluate the Failure Rate for Criterion 2. Again, online-LiRA and A-LiRA perform similarly, while offline-LiRA performs worse. However, as noted in Table 1, offline-LiRA is significantly more efficient than both online-LiRA and A-LiRA, making it a viable alternative when efficiency is the primary concern for model builders.

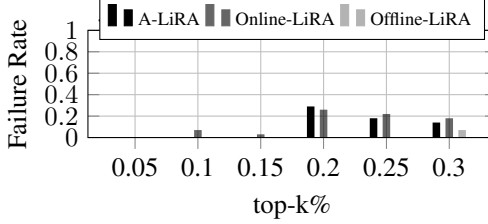
(a) Unlearn top-k%-risk samples of CIFAR-10.

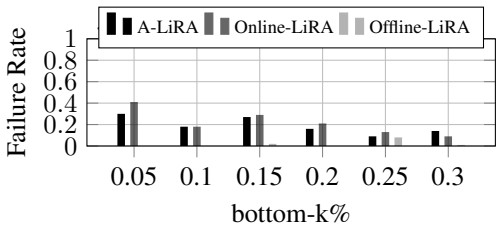
(b) Unlearn bottom-k%-risk samples of CIFAR-10.

Figure 7: Criterion 1: A-LiRA and online-LiRA detect more samples that failed to meet Criterion 1.

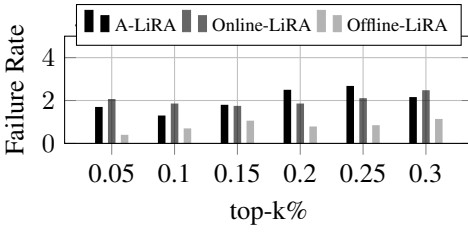
(a) Unlearn top-k%-risk samples of CIFAR-10.

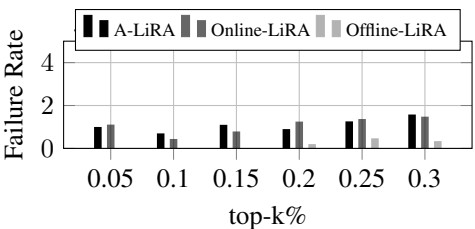
(b) Unlearn bottom-k%-risk samples of CIFAR-10.

Figure 8: Criterion 2: A-LiRA and online-LiRA detect more samples that failed to meet Criterion 2.

