# OpenReview forum: "Auditing Privacy Protection of Machine Unlearning"
_ICLR.cc/2025/Conference — ICLR 2025 Conference Withdrawn Submission_

### Official Review · Reviewer_BKw1 · 2024-10-24

**Soundness:** 2
**Presentation:** 2
**Contribution:** 2
**Rating:** 3
**Confidence:** 5

**Summary:**

The paper aims to audit the privacy protection of machine unlearning. The research topic and problem are interesting and hot. However, the threat model is not reasonable, and the experiments have not been compared with the latest auditing methods, which are not sufficient. Some detailed comments are as follows.

**Strengths:**

1. The structure of the paper is clear, making the paper easy to follow.
2. The figures in the experiments are good and clear.

**Weaknesses:**

1. The problem is the privacy protection of machine unlearning, but the main contribution is an attack method, which seems paradoxical to the problem. From the reviewer's view,  if we treat the proposed attack as the main contribution, it is better to focus only on the attacking problem. If we focus on the auditing problem, it would be redundant to introduce attacking and defense.

2. The threat model, the attacker has the white-box access to the and dataset is not reasonable. The attacker already knows the dataset and unlearned dataset, which means the attacker already knows the membership, why do we need additionally conduct membership inference attack?

3. As an auditing method for unlearning, the paper has not compared it with the latest MIA evaluation methods, such as [1][2][3]. The experiments only focused on the proposed method itself.

4. The experimental datasets are not sufficient, only CIFAR-10 and CIFAR-100. For image datasets, it is better to conduct experiments on STL-10 and imageNet.

5. For auditing the privacy protection of unlearning, only evaluating MIA based on unlearned models is not enough, as there are many privacy threats that are based on model differences before and after unlearning [4][5]. Even if the MIA metric of erased samples performs well on the unlearned models, the privacy is still leaked from the difference if MIA is high on the original model.

[1]. Hu, Hongsheng, et al. "Membership inference via backdooring." IJCAI-2022

[2]. Guo, Yu, et al. "Verifying in the dark: Verifiable machine unlearning by using invisible backdoor triggers." IEEE Transactions on Information Forensics and Security (2023).

[3]. Kurmanji, Meghdad, et al. "Towards unbounded machine unlearning." Advances in neural information processing systems 36 (2024).

[4]. Chen, Min, et al. "When machine unlearning jeopardizes privacy." Proceedings of the 2021 ACM SIGSAC conference on computer and communications security. 2021.

[5]. Hu, Hongsheng, et al. "Learn what you want to unlearn: Unlearning inversion attacks against machine unlearning." IEEE Symposium on Security and Privacy 2024

**Questions:**

No more questions.

---

### Official Review · Reviewer_Qz6c · 2024-10-30

**Soundness:** 3
**Presentation:** 2
**Contribution:** 2
**Rating:** 6
**Confidence:** 3

**Summary:**

This paper audits privacy risks of different machine unlearning methods for both unlearned and retained data samples, and introduces the novel membership inference attack A-LiRA, a more time-efficient variant of the state-of-the art likelihood ratio attack LiRA.

**Strengths:**

The paper is well-motivated and, for the most part, clearly written. It offers an interesting contribution to the field of privacy auditing and machine unlearning.

**Weaknesses:**

The work is missing an explicit discussion of its limitations (see Questions below).

The paper contains grammatical and typing errors. For example, a missing space in Line 97, a missing Section number in line 98, two verbs in a sentence + missing comma in line 148 and 152, etc.

**Questions:**

1) How likely is it that the assumption that in- and out-training measures are normally distributed holds, and what happens if it does not?
2) Does your work have any implications for adversaries with black-box access?

---

### Official Review · Reviewer_Rz9M · 2024-11-01

**Soundness:** 3
**Presentation:** 2
**Contribution:** 3
**Rating:** 5
**Confidence:** 3

**Summary:**

In this work, the authors study the privacy implications of unlearning methods. They provide two criterion for a correct privacy-preserving unlearning system, 1) that the unlearned samples should have more privacy in the unlearned model than the base model, and 2) that the retained samples should not degrade in terms of privacy after unlearning. In addition to providing these criterion, the authors develop a new method for efficient membership-inference attack, called A-LiRA. They provide experiments demonstrating the efficacy of A-LiRA and then empirically demonstrate that existing unlearning methods fail to satisfy their two criterion.

Overall, I think this paper provides good contributions, provides a practical speed-up to existing LiRA attacks, and shows some interesting empirical results. With the proper changes and answers to my questions, I am willing to move my score up to an "accept". I would love to see some more discussion of the "test" for Criterion 1 and some intuition for why it makes sense. As I wrote this review, I made myself more confused trying to understand it.

**Strengths:**

* This paper engages in discussion of a important, and often overlooked factor in unlearning research: are these methods actually "private?"
 * The authors also extend discussion to preserving privacy on other samples if the model was designed to be originally private.
 * A-LiRA is a neat approach to achieving efficiency gains and avoid the retraining of hundreds of shadow models.
 * The results demonstrate interesting phenomenon and provide new avenues of research for the field of unlearning, specifically in lines 494-497.

**Weaknesses:**

Weaknesses:
 * In unlearning there is a body of work that does focus on providing $\epsilon, \delta$ guarantees in the spirit of differential privacy and some look at unlearning impacts on other areas of trustworthy AI in addition to privacy. Given the goals of this work, you should definitely add discussion of them in your related work [1], [2], [3], [4]. Guo et al. even provides a direct link between unlearning and privacy guarantees.
 * Figure 1 doesn't seem to say anything. It seems like a generic table with no caption and it is really unclear what this contributes to the work. It does not refer to Criterion 1 or 2, nor does it refer to Equations 1,2, or 3.
 * How do you arrive at the $\epsilon$ upper bound in Eqn. 3? We have that $ln(TPR/FPR) < \epsilon$ only for the base model M, but not for the unlearning algorithm U. Mathematically this doesn't seem clear to me.
 * In your experimental results, I would like to see some baseline privacy violations in addition to k% plots. Specifically, for Fig. 2, how detectable are the samples before unlearning? In Fig. 4, same thing. In Fig. 5, what is the max(privacy violation) that you use from Eqn. 2?

Typos:
 * Line 098, you didn't finish the paragraph.
 * Line 148 "on D or D\x fits" > "on D or D\x and fits"
 * Line 152-153, "of the target model is not trained on the target model" > should this be "target model is not trained on the target sample"? Overall this entire paragraph was difficult to read, and I would recommend rewriting to improve its clarity.
 * Caption of Table 1: "classificaiton" > "classification"
 * Line 230, "Both Equation 1 and 3." - Equation 3 hasn't been introduced yet did you mean Eqn. 2? otherwise you should introduce Equation 3 before referring to it.
 * In line 254, should the range be $[0, \epsilon - max(ln (TPR/FPR))]$?


[1] Guo, Chuan, et al. "Certified data removal from machine learning models." arXiv preprint arXiv:1911.03030 (2019).
[2] Izzo, Zachary, et al. "Approximate data deletion from machine learning models."AISTATS 2021
[3] Neel, Seth, Aaron Roth, and Saeed Sharifi-Malvajerdi. "Descent-to-delete: Gradient-based methods for machine unlearning."  Algorithmic Learning Theory, 2021.
[4] Oesterling, Alex, et al. "Fair machine unlearning: Data removal while mitigating disparities." AISTATS 2024.

**Questions:**

* Notation-wise, is $PDF(\max\{\phi(l_{i,y})\} | \mathcal{N}(\mu, \sigma))$ supposed to mean the likelihood of $\max\{\phi(l_{i,y})\}$ under the specific Normal distribution (either in or out?) It was a little confusing to read PDF(* | normal).
 * In your test of Criterion 1, what if the inference attack is noisy? Then we expect that there will be some
 * At times I was confused by the tests for Criterion 1 and 2. I think in lines 341-346 you could elaborate a little bit more intuitively on each attack. For example for Criterion 1, it seems as though our goal is to test if a sample isn't detectable by MIA (because our ground truth is "non-member"). How can we ever expect to detect something we don't expect to be there? Morally, this test confuses me. A higher value means that we are correctly identifying samples as non-member (TPR/FPR), but why does this matter from an adversary's perspective, and if we can't identify samples as non-member, wouldn't that mean that they are actually not being unlearned properly? (as they are more likely to be detectable in the unlearned model) Criterion 2 makes a little more sense because the goal is to reduce our ability to detect if retained samples were used in training which is a more traditional privacy goal.
 * Is $\tau$ recomputed for each sample?
 * Do you think training with augmentations is part of the reason why A-LiRA works, as augmentations are in-distribution for the target model? Do you think A-LiRA would work in other settings without augmentation?
 * In Fig. 3, are you unlearning with a method or retraining? If retraining, I would change your captions.

---

### Official Review · Reviewer_25E6 · 2024-11-03

**Soundness:** 3
**Presentation:** 4
**Contribution:** 3
**Rating:** 6
**Confidence:** 4

**Summary:**

This paper aims to audit privacy unlearning through MIA, particularly using A-LiRA as the primary tool. It has two main contributions:

1. A-LiRA is a lightweight proxy of LiRA by deploying the augmentation information to fit the required IN and OUT distribution for distinguishing members and non-members.

2.  Two criterions are defined to characterize the success rate of machine learning, instead of focusing only on the sample to be forgotten, the authors argues that it is also important to ensure the remaining samples' privacy risks do not increase significantly due to unlearning.

With A-LiRA, the authors tested the performance of existing machine unlearning algorithms using ResNet-18 models trained on CIFAR-10 and CIFAR-1000 datasets, respectively. They found that most machine unlearning approaches, especially the approximate unlearning approaches,  produce suboptimal performances for both unlearned samples and the remaining samples.

**Strengths:**

1. The A-LiRA itself is interesting and has stand-alone contribution for MIA as it can significantly reduce the training time of LiRA, which is the main bottleneck. Though I donot know how is the reported performances generalize to various datasets, model architectures etc.

2. The paper verifies privacy onion effect in machine unlearning, this is interesting and poses new challenge to address the right to be forgotten.

**Weaknesses:**

see questions below

**Questions:**

1. Does A-LiRA demonstrate comparable performance to LiRA across various model architectures and datasets?

2. How does the parameter m affect performance? I assume that A-LiRA might be more sensitive to m (preferably larger) than LiRA, as A-LiRA relies on only one IN and OUT model.

3. In criteria one and two, there are thresholds t1 and t2 used for defining the acceptance matrix. What are the values of these thresholds in your experiments?

4. Regarding augmentation methods, is A-LiRA sensitive to the type of augmentation applied? Would standard augmentation methods such as flipping and rotation be more effective, or would advanced techniques like mixup and cutout yield better results?

5. I find it slightly distracting to propose a new MIA within this paper, as it occupies significant space, leading to an imbalance in the discussion of the machine unlearning aspect. For instance, it would be beneficial to explore how to address the challenge of the "privacy onion effect." Additionally, the discussions on exact unlearning and approximate unlearning methods seem brief, contributing to this imbalance.

6. Concerning the motivation, while it is interesting to observe the onion effect for the remaining samples, it also seems reasonable that if the unlearned sample is excluded from the dataset, some samples will inherently carry higher privacy risks than others. This is expected and may be fundamentally challenging to resolve. Why is it necessary to ensure the remaining samples' privacy risk to be smaller than the greatest privacy risk in the original model? Can it be possibly addressed?

7. Is focusing on the privacy risks of the remaining samples a novel contribution of your work, or has this topic been previously addressed in machine unlearning literature?

---

### Official Review · Reviewer_Y4Z7 · 2024-11-03

**Soundness:** 2
**Presentation:** 1
**Contribution:** 1
**Rating:** 1
**Confidence:** 4

**Summary:**

The authors propose privacy metrics for auditing the effectiveness of machine unlearning algorithms by way of membership inference attack (MIA), and propose a new MIA variant for LiRA that allows effective auditing.

**Strengths:**

Both machine unlearning and MIA are important topics to study in this field.

**Weaknesses:**

This paper is composed of two parts, proposing metrics for auditing the effectiveness of unlearning algorithms by MIA and proposing a new MIA solution. Both parts have limited contributions and technical depth.

1. The proposed metrics are pretty straightforward, and I do not think that they deserve that much space in this paper. Both Eq.(1) and (2) should be presented straightforwardly without the relaxations. Since there may not be an upper bound in the relaxation error in the first place (this is a more difficult problem than auditing I think), the authors may just use the error as the metric, rather than specifying them as criteria that some properties must hold.

2. The proposed metrics themselves do not have contributions to this field (MIA or unlearning). A good metric should do some of the following: enable new ways of looking at the problem, do the proposed metrics enable or improve the theoretical understanding, do they help design different frameworks for experiments, or do they inspire new solutions (e.g., attacks or unlearning algorithms)? From my understanding, the proposed metrics achieve none of the above requirements. Hence, these metrics are not interesting, nor should they be considered contributions to this field.

3. The proposed solution is unclear to me. What do the authors mean by ‘the transformed confidence values should vary less when the model is trained on the target data compared to those when the model is not trained on the target data’? How the variance is measured is not formally defined in the paper. Hence, I could not understand the intuition of the proposed attack.

4. Another point that does not make sense to me is that random data augmentation (flipping, shifting, rotating) could produce data points that are very similar to the original target data point. So, even if the target data point is not included in the target model, it is still very likely that the target model learns something particular to the target data point, through its augmented data. Why would the attack work in such scenarios, since both the in-shadow model and the out-shadow model contain the sensitive information of the target data point? Furthermore, when the number of augmented data is large enough, we can say that the original data itself causes a negligible impact on the in/out-shadow models (as it could be hidden in the crowd formed by its augmented counterparts). Then both the means and variances for the in-shadow and out-shadow models should be very similar, leading to very small likelihood ratios. Again, the attack could not work in this case.

5. Many related works are missing from both the unlearning and MIA fields, for example, Forget Unlearning: Towards True Data-Deletion in Machine Learning by Chourasia and Shah, and Low-Cost High-Power Membership Inference Attacks by Zarifzadeh et al. In particular, the latter one should be considered as a baseline for MIA, which is missed from the experiment section.

6. The experiment section is also limited, and the setup is not clear. Only LiRA is considered as the competitor and only TPR at 1% FPR is used as the evaluation metric (more FPRs should be used). Besides, just from AUC, the proposed solution is not better than LiRA (see Table 1). I do understand that the proposed solution is much more efficient. But is that the right direction to go? If machine unlearning does not happen too often or machine unlearning costs much more computation than privacy auditing, then one can certainly use the offline LiRA to achieve better MIA performance and better auditing results. Only one model (ResNet 18) is attacked, and the authors considered a specific training algorithm.

7. Why does unlearning worsen the privacy guarantee of DPSGD? DP is preserved under post-processing.

8. The proposed solution seems pretty restricted–random flipping, shifting, and rotating data applies to images only and cannot be extended to other data types, e.g., an image database where each record is composed of an image, the ID, the place and time the image was taken, etc. I could not see how a standalone image (with no identities linked) could pose privacy risks (if they are obtained by the model provider in the first place).

**Questions:**

See the above weaknesses.

---

### Note · Authors · 2024-11-13

I have read and agree with the venue's withdrawal policy on behalf of myself and my co-authors.